# Diabetic Retinopathy Screening and Registration in Europe—Narrative Review

**DOI:** 10.3390/healthcare9060745

**Published:** 2021-06-17

**Authors:** Elitsa Hristova, Darina Koseva, Zornitsa Zlatarova, Klara Dokova

**Affiliations:** 1Department of Physiotherapy, Rehabilitation, Thalassotherapy and Occupational Diseases, Training Sector of Optometry, Faculty of Public Health, Medical University of Varna, 9000 Varna, Bulgaria; zzlatarova@abv.bg; 2Department of Ophthalmology and Visual Sciences, Faculty of Medicine, Medical University of Varna, 9000 Varna, Bulgaria; darikoseva@gmail.com; 3Department of Social Medicine and Health Care Organization, Faculty of Public Health, Medical University of Varna, 9000 Varna, Bulgaria; klaradokova@gmail.com

**Keywords:** diabetic retinopathy, population screening, registry

## Abstract

Diabetic retinopathy (DR) is a leading cause of preventable vision impairment and blindness in the European Region. Despite the fact that almost all European countries have some kind of prophylactic eye examination for people with diabetes, the examinations are not properly arranged and are not organized according to the principles of screening in medicine. In 2021, the current COVID-19 pandemic moved telemedicine to the forefront healthcare services. Due to that, a lot more patients could benefit from comfortable and faster access to ophthalmology specialist care. This study aimed to conduct a narrative literature review on current DR screening programs and registries in the European Union for the last 20 years. With the implementation of telemedicine in daily medical practice, performing screening programs became much more attainable. Remote assessment of retinal pictures simultaneously saves countries time, money, and other resources.

## 1. Introduction

Diabetes mellitus (DM) has constantly increasing prevalence and incidence rates globally. According to the World Health Organization, about 422 million people worldwide have diabetes, and most of them live in low- or middle-income countries [1]. For people living with the disease, financially affordable diagnostic and treatment services are crucially important for their timely diagnosis, survival, and future quality of life.

Worldwide, over 95 million diabetic patients have DR, of which a third have vision-threatening DR and 7.6% macular edema [2]. Diabetic retinopathy (DR) is a leading cause of preventable vision impairment and blindness in the European Region as well [3,4]. The damaging effects of diabetes on vision could be prevented by screening programs based on early detection and prompt treatment [4,5].

The current technological advances in ophthalmology and the COVID-19 pandemic made telemedicine more accessible and a preferred way of providing health care services [6]. More patients benefit from comfortable and faster access to tele-ophthalmology specialist care. Increasing home-office time is one of the reasons more diabetic patients are informed, registered, and followed up. They spend more time on the internet searching, reading e-mails, and participating in online meetings related to their condition. Modern technologies deliver detailed, real-time videos, which when combined with accessible use of ophthalmology equipment allows more patients to a receive specialized medical opinion without even meeting an ophthalmologist.

Despite the fact that almost all European countries offer some kind of prophylactic eye examination for people with diabetes, the examinations are not always organized according to the principles of population-based screening established by Wilson and Jungner in 1968 and accepted by the World Health Organization (WHO) [7]. A well-developed screening program has the potential to identify early changes in the retina, allowing treatment to be offered before vision impairment or blindness appear.

Screening programs for DR are continuously changing and improving. New registries are created, and old ones are updated with increased coverage, new screening methods, and sources of information. The last review article for DR screening programs in Europe dates from 2015 by Pieczynski et al. [8]. This is a long period for a constantly developing highly technological specialty such as ophthalmology, which is our main reason for conducting this narrative review.

Early diagnosis of DR may save a substantial amount of resources for treatment and may increase the patients’ quality of life. One of the basic problems in screening for diabetic retinopathy is that many countries cannot detect everyone who has a diagnosis of diabetes. Without a proper registry of diabetic patients, some people may never be screened for DR.

This study aimed to conduct a narrative literature review on current DR screening programs and registries in the European Union (EU) member states for the last 20 years. Their geographical area, age group coverage, type of screening methods, time period, etc. are presented.

## 2. Materials and Methods

A literature search of the PubMed and Google Scholar databases (January 2000 to January 2020) was conducted using the following terms and text words: (“diabetic” OR “retinopathy” OR “screening” OR “registry”) combined with the name of each country member of the EU. Additionally, article bibliographies and grey literature sources were searched for relevant publications. Publications in English language, accessible in full text were selected for the review. Since the United Kingdom (UK) left the EU on 1 February 2020, all UK countries are included in this review.

### Eligibility Criteria

Studies were included based on the following criteria: (1) provide information on a recent screening program or functioning DR registry; (2) have a study population consisting of individuals living in any one of the 28 country members of the European Union by the date of the last literature search (January 2020); (3) be written in the English language and published in a peer-reviewed journal.

Unrelated and duplicate publications were excluded after an initial review of the titles and abstracts. The process of study selection, including identification, screening, eligibility, and inclusion is illustrated in a flowchart presented in Figure 1. The review complies with the recommendations of the Scale for the Assessment of Narrative Review Articles (SANTRA) [9].

The checklist for the systematization of studies on information availability, quality of the articles, and methodology of populational DR registers included:Country/Region/City of the screening program/registryFirst authorYear of publicationAim of the articleName of screening program/registerTime period of register/screening functioningDefined geographical area coveredTarget population in which cases ariseNumber of the population from which DR cases are identified (where indicated)Age group of target populationSources of information for cases of diabetes (where indicated)Screening method

## 3. Results

We identified 160 articles with the keywords “diabetic”, “retinopathy”, “screening”, and “registry” as well as 270 using “diabetic”, “retinopathy”, “registry”. The full texts of the 44 remaining articles were reviewed after we were sure that the studies met inclusion criteria. Summary characteristics of the diabetic screening programs or registries in the European Union from each of the included studies are presented in Table 1 and the following text.

Only a few EU countries (Finland, Sweden, Ireland, and Denmark) have national DR screening programs (Figure 2). These programs are predominantly based on preexisting national population-based diabetes registers.

In Finland, the existing Finnish Register of Visual Impairment (RVI) does not require the permission of the patient for registration. Every patient with a diagnosed visual impairment is entitled to an ophthalmology examination, and patients with DR are included by the ophthalmologist in the register. The method of the screening is not specified, as every ophthalmologist might use a different approach. The organization guarantees that almost all diabetic patients are screened, although there is no particular DR register in the country [10].

In Sweden, unlike other EU countries, the Swedish Diabetes Registry is the source of information for the DR screening of patients with type 2 diabetes and type 1 diabetes who are on insulin treatment and for patients with a diagnosis at 30 years of age or younger. The register includes children and adults with a diagnosis of diabetes for up to five years when first recorded in the registry. Of the 251,386 diabetic patients for 2008 registered at a primary healthcare center and living in the catchment area of the eye clinic of Linköping University Hospital, 3515 were diagnosed with DR [22]. Similar to Finland, the method of screening is not indicated. Both countries have local screening programs along with national ones.

In Ireland, there is a diabetes register covering the Mid-West Ireland territory. Provisional lists of diabetic patients are made by a combination of Primary Care Reimbursement Service (PCRS) sources, Hospital In-Patient Enquiry (HIPE) scheme, Patient Administration System (PAS), and Laboratory Information System (LIS) sources [16]. The screening methods include a dilated ocular examination by a single ophthalmologist, primarily with slit-lamp biomicroscopy or with indirect and direct ophthalmoscopy in cases of poor patient mobility. No patient was screened with retinal photography. Of the 1943 patients who were offered screening during the study period, 1434 attended, and 405 were diagnosed with different stages of DR. Based on the lists, the Mid-West diabetes register of the Diabetic Retina Screen national program was later established. The program has its own website, which can be used by every patient for self-registration. GPs can register patients, and the program constantly receives the contact details of the majority of persons with diabetes from the National Health Schemes. Two photographs of each eye with a digital camera are used as the screening method. According to the results evaluated by an ophthalmologist, every patient receives a notification letter for the next appointment [23].

Denmark has its Danish Registry of Diabetic Retinopathy (DiaBase) covering the whole country. DiaBase contains data collected from hospital eye departments and private ophthalmological practices for all diabetes patients aged ≥18 years in the five regions of Denmark. Data from each screening visit are collected and sent electronically to DiaBase. The screening method includes slit lamp or photography-based examination according to ophthalmologist personal preference. Prevalence of non-proliferative diabetic retinopathy (early stage of DR in which symptoms are mild or nonexistent) of 14,034 patients and proliferative DR (advanced stage of diabetic eye disease which affects central and peripheral vision) of 3118 patients among a total of 77,968 diabetes patients nationwide for a two-year period have been reported. The proportion of patients with diabetic retinopathy regression in Denmark is greater than the proportion of patients with progression, although the number of newly diagnosed patients with diabetes has been increasing every year [18]. This fact demonstrates the success of DiaBase and the importance of the existing well-functioning national registry and screening program.

In France, a regional telemedical screening network (OPHDIAT) [13] was initially established in a Parisian hospital. A non-mydriatic camera installed in the diabetology department has been used for fundus photographs of diabetic patients, which have been taken by trained nurses and electronically transmitted to the ophthalmology department for interpretation. Only patients with no documented DR or mild DR are referred for screening. Sixteen screening centers were progressively opened in Paris and the entire surrounding area, linked through a central server to an ophthalmologic reading center [14]. The centers are located in the diabetology departments of 11 different hospitals and healthcare centers. Two retinal photographs are generally taken by trained orthoptists and nurses, one centered on the posterior pole and the other on the optic disc. Photos are sent together with the relevant clinical information to the central medical server for interpretation and storage.

In Germany, there are no existing screening programs or DR registry. Information about DR patients comes from a population-based registry of blindness in the country. The register contains a code indicating the probable cause of blindness. That coding is based on a review of written medical and ophthalmological opinions. Some 589 people have been registered as blind due to diabetes, but no exam was performed on the 4373 blind people in Germany in the years 1990–1991 [24].

In Italy, there are no national data about legal blindness due to DR, and there is no national registry of patients with DM. However, few geographically limited population-based studies report the prevalence and incidence of DR in the past [25,26]. There are several telemedicine regional screening programs in progress [27,28,29]. All of them are based on fundus photos taken in diabetes centers by trained nurses, paramedical staff, or other medical personnel. The preferred screening method is nonmydriatic color fundus photos electronically transmitted to a reading center for examination by certified expert ophthalmologists.

The United Kingdom is the largest country in Europe with nationwide screening for all patients with DM above the age of 12 years. Screening started in 2003, reaching national coverage in 2008 [30]. The Health Improvement Network (THIN) database was created for the whole UK territory [31]. It contains over 9 million patients in total, which covers around 6% of the UK population. The THIN database contains individual patient information recorded by primary care practitioners as part of their routine clinical care. There is no separate registry of patients with DR, but it is an enormous database containing information about all diabetic patients and patients with retinopathy related to diabetes. The UK comprises four constituent and relatively autonomous countries: England, Scotland, Wales, and Northern Ireland.

England has a National Screening Program for diabetic retinopathy. Everyone who has a Primary Care Physician and a respective NHS identifier number and has been diagnosed with diabetes receives an invitation letter for a diabetic eye screening appointment once a year. The rate of retinopathy is 2807 per 100,000 screened in 2003–2016 year. Screening includes two 45° fields mydriatic photographs [32]. Screening images are evaluated by higher-ranked graders with a subsequent referral to an NHS hospital with eye services when that is indicated.

Scotland also has its national screening program [20]. In 2006, Scotland launched a national Scottish DR Screening service (DRS). Screening aims to prevent vision loss due to proliferative DR (PDR). Patients identified as high risk are referred to eye clinics for assessment and further management. Patients eligible for screening are identified via the national diabetes registry—the Scottish Care Information-Diabetes Collaboration (SCI-DC) database. The screening examination includes a single central 45° field digital photograph. Mydriasis is performed only if poor-quality images are obtained. When photographic images are ungradable, a slit-lamp examination is undertaken.

In Wales, a population-based diabetes register was established based on information from the hospital patient administration systems and lists of diabetic patients from all primary care practices in 2001. The screening is undertaken photographically using a digital camera attached to a fundus camera in a mobile unit. It is performed by optometrists trained to use slit-lamp biomicroscopy and non-contact lens or by ophthalmology nurses who are trained to take the photographs and grade them. Later on, in 2005, the Diabetic Retinopathy Screening Service for Wales was initiated as a community-based mobile screening service [21]. All persons invited for screening receive an appointment letter with a date, time, and venue for the screening. Thirty photographic teams consisting of a healthcare professional and an accredited photographer take two 45° fields (one macula centered and one nasal) digital fundal photographs following mydriasis. Images are graded by accredited retinal specialists.

Northern Ireland has a regional Diabetic Eye Screening Program (DESP). Invitation letters with a fixed date and time for screening are sent to all eligible people. Two or more photographs are taken of each eye using a special camera by an optometrist. If a person is over 50 years of age, eye drops are instilled about 15 min before the test to dilate their pupils [33].

## 4. Discussion

Diabetic retinopathy is an eye disease of great social importance as a leading cause of blindness in working-aged people. Early diagnosis and timely initiated treatment are crucial for saving the sight and the quality of life of diabetic patients. Many problems may delay diagnosis of diabetic retinopathy, including difficult access to an eye specialist and patients’ lack of sufficient information about the risk of vision loss caused by diabetic retinopathy. Systematic screening for DR has a potential to reduce disease burden and visual impairment.

The first diabetic retinopathy screening program was introduced in Iceland in 1980 [34]. For the past forty years, different screening programs have been adopted in Europe: hospital-based, regional, and national. Various types of diagnostic tests are used: direct or indirect ophthalmoscopy or fundus photography. Depending on the traditions of the healthcare system in each country, different medical specialists are engaged in the screening programs: ophthalmologists, optometrists, general practitioners, technicians, trained photograph readers.

The present literature review reveals that the UK national diabetic eye screening program is the largest national, well-organized existing program in Europe. Despite the free access to screening, more than one-third of patients with diabetes do not take advantage of the preventive service. While in the UK technicians perform one or two 45° fundus images, graded by trained graders, in Denmark, screening is performed by ophthalmologists using slit lamp indirect ophthalmoscopy or fundus photography. Denmark is the only country in EU that has a diabetic retinopathy registry (DiaBase). Some countries such as Sweden and Ireland have national screening programs based on a national diabetic registry, but they do not report maintaining a diabetic retinopathy registry with systematically collected information about patients’ retinopathy.

In Norway, a European country although not a member of the EU, two registers—The Norwegian Diabetes Register for Adults and The Norwegian Childhood Diabetes Registry—serve as the basis for the national diabetic retinopathy screening program [35]. The DR diagnosis is made by ophthalmologists based on the results of retinal photographs and/or fundoscopy. Ophthalmologists report the results to the referring doctors who in turn report the results to the national DR register.

We could not find published information about national diabetic retinopathy screening programs or registries in Central and East European countries that are members of the EU. Pieczynski et al. [8] mention the existence of diabetic retinopathy screening programs in Hungary, Czech Republic, and Poland, but do not provide information about their organization. A few regional screening programs have been conducted in Bulgaria, involving eye examination with direct or indirect ophthalmoscopy by ophthalmologists. Vassileva et al. [36] screened 369 diabetic patients and estimated 28% incidence of DR. Two small-scale programs report 35% incidence of DR among diabetic patients [37] and 41% DR incidence among patients with poor control of their diabetic disease [38].

The introduction of new technology in ophthalmology allowed remote assessment of diabetic patients, simultaneously saving time, money, and other resources. Mansberger et al. compared the new DR screening with traditional retina examination [39], revealing that the new technology increases DR screening accessibility. They concluded that remote DR screening could be performed by primary care clinics.

A rising interest in autonomous artificial intelligence (AI)-based ophthalmological devices has been observed in recent years [40,41,42,43]. The first autonomous AI-device with FDA approval for diagnosing more than mild DR was a creation of Abramoff et al. with a reported sensitivity of 87% and specificity of 91% [44]. AI-screening can be used in large groups of people who otherwise might not be seen by an ophthalmologist in regular exams. In the current era of smart technologies, screening cameras integrated into portable digital devices are not too far off in the future. Such a procedure will be time- and money-saving for patients. Until that moment becomes a reality, the screening process will be based on fundus photos performed by trained healthcare professionals who are not ophthalmologists, most often nurses, optometrists, etc.

In 2017, the International Council of Ophthalmology (ICO) published updated guidelines for diabetic retinopathy screening and diabetic eye care specific to resource settings. Using these guidelines, each country, according to its healthcare system, financial resources, and available medical specialists, has to create and implement a national DR screening program, which has the potential of significantly reducing blindness due to DR.

## 5. Conclusions

In the era of technology and preventive medicine, there are still developed European countries with no national DR screening programs or registry. Based on our literature review, countries of Western Europe have successful ongoing screening programs. For Eastern European countries, there is insufficient information. In the meantime, global incidence rates of DM rise steeply due to aging populations.

## Figures and Tables

**Figure 1 healthcare-09-00745-f001:**
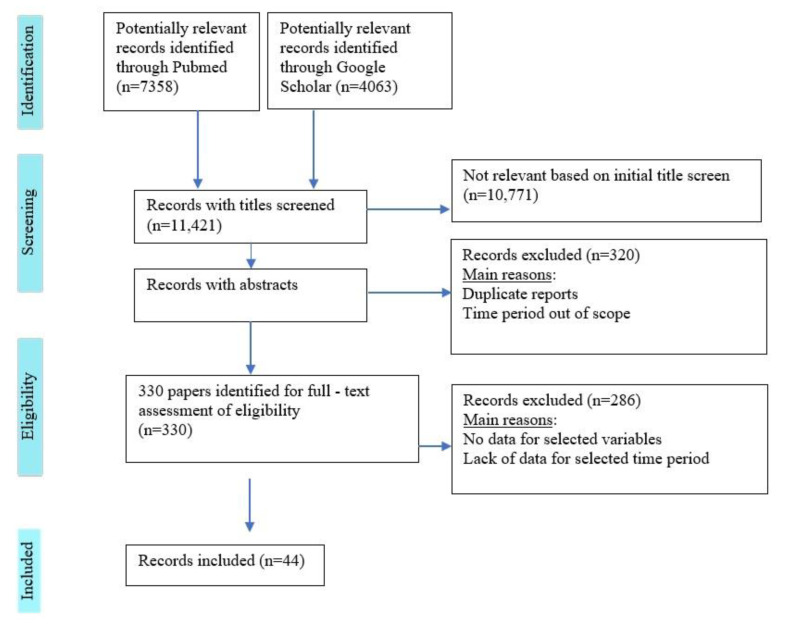
Flow diagram describing the systematic search for publications on screening programs and registries for DR in EU countries.

**Figure 2 healthcare-09-00745-f002:**
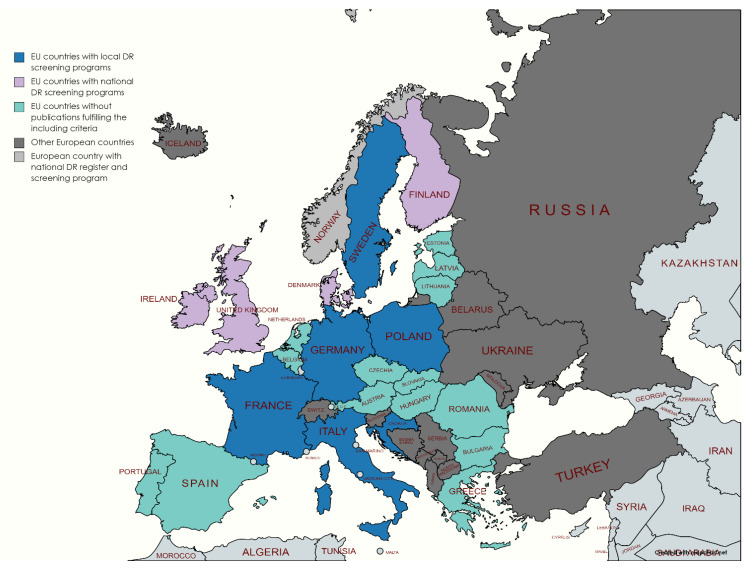
Geographical map of European countries based on presence of DR screening programs and registries.

**Table 1 healthcare-09-00745-t001:** Summary characteristics of the included studies.

Country	First Author and Year of Publication	Time Period	Covered Geographical Area	Age Group Screened (Years)	Number of Participants/DR Relative Share	Name of Register/ Type of Study
Finland	Laitinen et al. 2010 [10]	2000–2001	Whole of Finland	≥30	7413/1%	Cross-sectional nationwide population survey
	Hautala et al. 2014 [11]	2007–2011	35 municipalities of the Northern Ostrobothnia Hospital District	No age limitation	14,866/23% mild background retinopathy, 31% moderate or severe background retinopathy, 3%—PDR	Finnish Register of Visual Impairment
	Laatikainen et al. 2016 [12]	1982–2010	Whole of Finland	18–64	42,626/0.09%	Finnish Register of Visual Impairment
France	Massin et al. 2008 [13]	2004–2006	Paris region	1–106	15,307/23.4% DR	Regional telemedical network/OPHDIAT/
	Schulze-Döbold et al. 2012 [14]	2004–2009	Paris region	All ages	38,596/14.7% advanced stage retinopathy	Regional telemedical network/OPHDIAT/
Ireland	Kelliher et al. 2006 [15]	1996–2003	Whole of Ireland	No age limitation	470/7% DR	The National Council for the Blind in Ireland (NCBI)
	James et al. 2016 [16]	2010–2012	Mid-West of Ireland	≥20	1434/20.1% background retinopathy, 8.2% sight threatening retinopathy	Mid-West Diabetic Retinopathy Screening Pro- gramme (MWDRS)
	Tracey et al. 2016 [17]	2004–2013	Whole of Ireland	18–69	57,626–109,842/0.4–1.9% during 10-year period	National Council for the Blind of Ireland
Denmark	Andersen et al. 2016 [18]	2014–2015	Whole of Denmark	≥18 years	77,968/18% NPDR, 4% PDR	Danish Registry of Diabetic Retinopathy (DiaBase)
	Holm et al. 2018 [19]	2010 and still ongoing	Copenhagen City	No age limitation	21,000/11.3% DR	National Patient Register Danish Adult Diabetes Database
Scotland	Looker et al. 2014 [20]	2006–2010	>99% of the Scottish population	≥12	187,822/0.6–1.8% referable background retinopathy	National diabetes registry—Scottish Care Information-Diabetes Collaboration (SCI-DC) database
Wales	Thomas et al. 2015 [21]	2005 and still ongoing	Whole of Wales	≥12	91,393/32.4% DR	Diabetic Retinopathy Screening Service for Wales (DRSSW)

## Data Availability

Not applicable.

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
