# Peer review of "Diabetic Retinopathy Screening and Registration in Europe—Narrative Review"

_healthcare, 2021, doi:10.3390/healthcare9060745_

Round 1

Reviewer 1 Report

In this study, the authors reviewed the current DR screening programs and registries in Europe, specifically in Finland, Sweden, Ireland, and Denmark, for the last 20 years. Since DR is a leading cause of preventable vision impairment and blindness worldwide, the finding may be potentially significant to the broader readers. The contents are well-organized and written, but the current manuscript needs several clarifications described below.

  1. Table 1 is informative but less organized. It would be better to separate results in each country to show the common challenges. 
  2. The authors discussed the rate of patients diagnosed with DR in each country by stating mostly numbers but some are expressed as percentages as in Denmark. It would be clearer to use either number or percentage or both so that readers can easily understand the difference between the countries.
  3. Similarly, in Table 1, it would be good to have the number of participants in each program so that readers can easily understand the size of the screenings.
  4. The authors used a technical term, for example, “non-proliferative diabetic retinopathy” and “laser-treated retinopathy”. It would be better to briefly explain those terms for readers who are not familiar with the current DR treatment.

Reviewer 2 Report

The manuscript “Diabetic Retinopathy Screening and Registration in Europe -  Narrative Review” by Hristova et al. tries to identify Diabetic Retinopathy Screening and Registration programs, which exist in Europe.

The topic can be of interest to some healthcare policymakers; however, it will not be of interest to a broad audience in the current form.

The work has significant design flaws, so I would recommend rejecting the paper in the current form. The basis for my recommendation is following:

  1. Methods are inadequate. Researching only PubMed is not sufficient
  2. Inclusion criteria include ambiguous criteria like “First Author,” etc., thus the study cannot be verified or reproduced.
  3. The geographic area (Europe) is too broad and includes a wide range of socio-economic regions, including Russia. Such selection is not justified
  4. Discussion and Conclusions are almost non-existent
  5. The article is sprinkled with buzzwords (telemedicine and AI); however, there is no logical connection between these concepts and the research question.

Reviewer 3 Report

Well written review which documents the international need for screening of diabetic retinopathy.Would benefit from a figure.I would suggest a map.

Round 2

Reviewer 1 Report

All the concerns in the previous review were appropriately addressed by the authors and the revised manuscript now reads very well. 

Author Response

Thank you very much for your second review.

Reviewer 2 Report

The revised version of the manuscript “Diabetic Retinopathy Screening and Registration in Europe -  Narrative Review” by Hristova et al. has been significantly improved in quality. The addition of Fig 1 and especially Fig 2 has dramatically improved the article.

In the current form, it can be of interest to a broader audience.

My primary concerns have been addressed in general. However, I still have several concerns regarding Google search

  1. Google search is not reproducible in general. The returned results will be based on cookies and previous search history. The only meaningful way to improve reproducibility is to use the Incognito tab.
  2. Google Scholar search returns a large number of articles. For example, when I entered “Diabetic Retinopathy Register Poland“ 2000-2019 as per the authors’ suggestion, I have received about 3,870 results (06 sec), which is on par with n=4063 mentioned by authors for all European countries. Did authors go through all 3,870 papers returned by Google, or they cherry-picked from several first pages?

Minor issues:

  1. Line 272: “Pieczynski J. and Grzybowski A.” is a pretty unusual way to cite an article. Consider using a standard approach.
